# Pulmonary Histoplasmosis: A Clinical Update

**DOI:** 10.3390/jof9020236

**Published:** 2023-02-10

**Authors:** Nicolas Barros, Joseph L. Wheat, Chadi Hage

**Affiliations:** 1Department of Medicine, Indiana University School of Medicine, Indianapolis, IN 46202, USA; 2Division of Infectious Diseases, Indiana University Health, Indianapolis, IN 46202, USA; 3MiraVista Diagnostics, 4705 Decatur Boulevard, Indianapolis, IN 46241, USA; 4Pulmonary Allergy and Critical Care Medicine, University of Pittsburgh, Lung Transplant, University of Pittsburgh Medical Center, Pittsburgh, PA 15213, USA

**Keywords:** histoplasmosis, pulmonary histoplasmosis

## Abstract

*Histoplasma capsulatum*, the etiological agent for histoplasmosis, is a dimorphic fungus that grows as a mold in the environment and as a yeast in human tissues. The areas of highest endemicity lie within the Mississippi and Ohio River Valleys of North America and parts of Central and South America. The most common clinical presentations include pulmonary histoplasmosis, which can resemble community-acquired pneumonia, tuberculosis, sarcoidosis, or malignancy; however, certain patients can develop mediastinal involvement or progression to disseminated disease. Understanding the epidemiology, pathology, clinical presentation, and diagnostic testing performance is pivotal for a successful diagnosis. While most immunocompetent patients with mild acute or subacute pulmonary histoplasmosis should receive therapy, all immunocompromised patients and those with chronic pulmonary disease or progressive disseminated disease should also receive therapy. Liposomal amphotericin B is the agent of choice for severe or disseminated disease, and itraconazole is recommended in milder cases or as “step-down” therapy after initial improvement with amphotericin B. In this review, we discuss the current epidemiology, pathology, diagnosis, clinical presentations, and management of pulmonary histoplasmosis.

## 1. Introduction

*Histoplasma* is a dimorphic fungus that grows as a mold in the environment and as a yeast in human tissues. Human histoplasmosis is caused by two distinct organisms: *Histoplasma capsulatum* var. *capsulatum* and *Histoplasma capsulatum* var. *duboisii*. While *H. capsulatum* var. *duboisii* occurs exclusively in Africa, *H. capsulatum* var. *capsulatum* has been identified on all continents except for Antarctica (Figure 1) [1]. In this study, we will describe the pulmonary syndromes associated with *H. capsulatum* var. *capsulatum*, which will be referred to as *Histoplasma capsulatum*.

In North America, the highest endemicity areas include the valleys of the Mississippi and Ohio Rivers in the central and eastern United States. In these areas, the incidence is estimated to be 6.1 cases per 100,000, and 80–90% of the population will be exposed to histoplasmosis during their lifetime [2,3]. In recent decades, environmental disruptions, climate change, increased travel and connectivity, and increased immunosuppressive conditions have led to changes in the epidemiology of histoplasmosis [1,4]. A recent study has shown a significant number of patients being diagnosed with histoplasmosis outside of the historical geographic distributions [5].

Histoplasmosis is also endemic to Central and South America, where the prevalence of infection may be higher than 30% [6]. There is a wide regional variability with seroprevalence of 0.1% in Chile, 20% in Peru, 35–40% in Argentina to almost 90% in certain areas of Brazil [7].

The advent of the HIV pandemic has highlighted the epidemiology of Histoplasmosis, as patients living with HIV are at an increased risk of having symptomatic disease [8]. The incidence of symptomatic histoplasmosis in Central and South America is unknown, as many cases are misdiagnosed as tuberculosis. However, modeling data suggests incidences of up to 1.5 cases per 100 people living with HIV [6]. This is particularly important as disseminated histoplasmosis occurs in approximately 2–5% of patients with advanced HIV, and in some countries, including French Guyana and Colombia, disseminated Histoplasmosis is the most common AIDS-defining illness. Improved access to active retroviral therapy invariably decreases the morbidity and mortality of histoplasmosis [9].

In China, 75% of the reported cases occurred within the Yangtze River basin. The prevalence of Histoplasmin skin test reactivity ranges from 6% to 50% [4]. Southeast Asia is also a hyperendemic area, with a prevalence of up to 33% [7].

While *Histoplasma* is broadly distributed around the globe, the lack of universal access to sensitive diagnostic tests likely lead to a gross underestimation of the true burden of disease.

## 2. Pathogenesis

*H. capsulatum* may be found in microfoci within endemic areas. These microfoci include soil enriched in nitrogen with pH levels between 5 and 10, which is usually due to large amounts of bird excreta or bat guano [9].

The mycelial phase in the environment has two types of conidia: macroconidia measuring 8–14 μm in diameter with distinct tuberculate projections on the surface (ship’s wheel) and microconidia measuring 2–5 μm in diameter (infectious form). Microconidia formed in the mold phase can easily be aerosolized by disruption of their microfoci by farming, remodeling or demolition of old buildings, clearing areas where blackbirds have roosted, or exposure to chicken coops and caves [1]. Infection occurs following the inhalation of microconidia, which can reach the lower respiratory tract. The temperature shift and changes in nutritional composition trigger the morphological transformation into yeast [10].

Once in the alveoli, *H. capsulatum* evades the soluble pattern-recognition receptors, collectins (surfactant protein A and surfactant protein D), and rapid internalization into macrophages. Yeasts can be internalized by other phagocytes, including dendritic cells, neutrophils, and natural killer cells, but are incapable of surviving within them [11].

The heat shock protein 60 (Hsp60) of *H. capsulatum* is the major surface adhesin to macrophages and binds to complement receptor 3 (CR3) [12]. Dectin-1 recognizes 1-3 β glucan which is a major polysaccharide in the cell wall of most fungal pathogens. Engagement of Dectin-1 leads to production of pro-inflammatory cytokines, including TNF-α and IL-6. *H. capsulatum* expresses an outer layer of 1-3 α glucan concealing the 1–3 β D glucans impairing the recognition by Dectin-1. In addition, the H antigen is a beta-glucosidase that remodels the yeast cell wall and is involved in nutrient acquisition [13]. Altogether, the Hsp60-CR3 engagement, while avoiding dectin-1, leads to yeast internalization without triggering pro-inflammatory cascades [11].

Once inside the macrophage, *Histoplasma* evades reactive oxygen and nitrogen species and is also able to regulate the phagosome acidification and the phagolysosomal fusion. The M antigen, also known as catalase B, counteracts the oxidative stress from the host’s defense mechanisms [13]. Within the macrophages, the yeast reproduces by budding into daughter yeast (narrow budding).

Macrophages are able to migrate to other areas of the body, and *H. capsulatum* uses them as a trojan horse for dissemination into other tissues. Subsequently, the yeast can induce host cell death, allowing them to migrate to another phagocyte [10].

Dendritic cells provide the link between the innate and adaptive immune responses by acting as antigen-presenting cells [14]. Human dendritic cells drive CD4+ and/or CD8+ T cell proliferation, which in turn can either clear the pathogen or lead to granuloma formation. CD4+ T cell polarization into Th1 phenotype leads to production of IFN-γ and TNF-α, which activates intracellular killing and is pivotal for control of the infection [15]. While Th17 responses are critical in multiple fungal infections, the role of these helper cells in histoplasmosis is not clearly elucidated [16].

Unlike Th1 and Th17, polarization into Th2 leads to increased levels of IL-4 and other type 2 cytokines, which in turn lead to alternatively activated macrophages (M2) [17]. Macrophages with the M2 phenotype are critical for tissue repair and fibrosis but are unable to kill intracellular pathogens [18].

While the vast majority of people exposed to *H. capsulatum* remain asymptomatic or have mild, self-resolving symptoms, some patients can develop several Histoplasmosis syndromes [1]. Ultimately, the outcome of infection and clinical presentations depends on the dynamic interactions between the host’s innate and adaptive immunity and the fungal virulence factors (Figure 2).

## 3. Diagnosis

The diagnosis of histoplasmosis can be ascertained by the direct observation or isolation of the pathogen (culture, histopathology, and cytopathology) or by the detection of antigens, antibodies, or nucleic acids. Each test has different performances that may vary according to the clinical syndrome and underlying host immune system (Table 1).

### 3.1. Culture, Histopathology, and Cytopathology

Isolation of *H. capsulatum* from clinical specimens remains the gold standard for the diagnosis of histoplasmosis, though histopathologic or direct microscopic identification are also considered definitive diagnoses [19].

Clinical samples sent to the laboratory for culture are plated on a nutrient-poor medium such as Sabouraud agar and incubated at 25 °C to 30 °C. Mycelial growth occurs within 2 to 3 weeks but may take up to 8 weeks. Depending on the maturity of the mycelia, microscopic examination with the lactophenol cotton blue test can show septated hyphae, microconidia, or macroconidia. The tuberculate projections on the surface (ship’s wheel) are highly suggestive of *Histoplasma capsulatum*, though they can also be noted on fungi of the genus *Sepedonium* [20]. A definitive diagnosis requires a confirmatory test. A chemiluminescent DNA probe (AccuProbe test; Gen-Probe Incorporated, San Diego, CA, USA, 2011) is commercially available [21]. More recently, a proteome-based technique, matrix-assisted laser desorption ionization-time of light mass spectrometry (MALDI-TOF MS), has been shown to be highly accurate for the identification of *H. capsulatum*. This test can identify yeast forms and early mycelial cultures [22].

The sensitivity of culture-based diagnosis relies on the specimen and burden of disease. The highest sensitivities are observed in blood cultures or bone marrow cultures in patients with disseminated disease and in BAL fluid or sputum in those with chronic pulmonary histoplasmosis, while the lowest are observed in acute and subacute pulmonary histoplasmosis. In addition, the cultures have higher sensitivity in immunocompromised patients [23].

A recent meta-analysis assessing different diagnostic assays for progressive disseminated histoplasmosis in patients with advanced HIV showed that the overall sensitivity for culture was 77% (95% CI 72–81%) with blood and bone marrow cultures having the highest sensitivities ranging from 60–90% and respiratory samples showing the lowest sensitivities ranging from 0% to 60% [24].

**Table 1 jof-09-00236-t001:** Sensitivity of diagnostic tests in different clinical forms. * Mediastinal forms do not have well established sensitivity for the different testing strategies. Table adapted from Hage Ca, et al. [25].

	Pulmonary			Mediastinal			
Method	Acute	Subacute	Chronic Cavitary	Adenitis *	Granuloma *	Fibrosis *	Progressive Disseminated
Antigen	83%	30%	88%	May be positive	Usually negative	Negative	92%
Antibody	64%	95%	83%	Usually positive	Usually positive	Usually positive	75%
Pathology	20%	42%	75%	May be positive	May be positive	Uncommonly positive	76%
Culture	42%	54%	67%	May be positive	Uncommonly positive	Negative	74%

Histopathology can show characteristic features including granulomas (both caseating or non-caseating), and staining with Grocott-Gomori’s methenamine silver or Wright’s periodic acid-Schiff allows for the visualization of ovoid, narrow-based budding yeasts. Yeasts are usually identified within macrophages but can occasionally be seen free in the tissue (Figure 3). Similar to culture-based diagnosis, the sensitivity of histopathological identification varies depending on the tissue sample and burden of disease, with higher sensitivities in disseminated and chronic pulmonary disease and lower sensitivities in acute and subacute pulmonary histoplasmosis [25,26].

Cytopathology allows for the examination of tissue aspirates and fluid, which is less invasive than histopathology. Most commonly, cytopathology is obtained from bronchoalveolar lavage, where it has a sensitivity of 48% for acute pulmonary histoplasmosis. However, when combined with antigen testing, the sensitivity rises to 97% [27].

### 3.2. Antigen Culture, Histopathology, and Cytopathology

The detection of *H. capsulatum* polysaccharide in urine and serum was first developed in 1986 as a sandwich radioimmunoassay and was subsequently reformulated into an enzyme immunoassay (EIA) in 1989. A second-generation EIA was introduced in 2004, which allowed for semiquantitative results and reduced the number of false-positive results caused by human antirabbit antibodies, and most recently, a third-generation quantitative test (MiraVista *H. capsulatum* Galactomannan EIA) was FDA-approved [20]. Another platform, using the analyte-specific reagent *H. capsulatum* antigen EIA (IMMUNOMYCOLOGICS, Norman, OK, USA) has been FDA cleared [28].

A large multicenter study evaluated the performance of antigen detection in patients with histoplasmosis. The sensitivity of the antigen testing was higher in immunocompromised patients (including advanced HIV) than in immunocompetent recipients. Furthermore, the test performance varied greatly depending on the severity of the disease. In patients with progressive disseminated disease, antigen in urine was detected in 92% of all patients (95% in advanced HIV, 93% in other immunocompromised states, and 73% in immunocompetent patients), 83% of patients with acute pulmonary disease, 30% in those with subacute disease, and 88% in those with chronic pulmonary disease [23]. Patients with mediastinal adenitis may have positive antigen tests, though patients with mediastinal granuloma or fibrosing mediastinitis are usually negative [29].

The performance of antigen detection is slightly more sensitive in urine than in serum, though combining both tests can increase the detection from 65% to 83% in acute pulmonary histoplasmosis [30].

Antigen testing can be completed in many body fluids. In a study of pulmonary histoplasmosis, antigen was detected in 93% of bronchoalveolar lavage (BAL) samples, compared with only 79% in urine and 65% in serum. Furthermore, combining BAL antigen and BAL cytopathology had a sensitivity of 97% [27]. Cerebrospinal fluid (CSF) testing was positive in 78% of the patients with CNS histoplasmosis and had a specificity of 97%. Combining the test with CSF immunoglobulin G anti-*histoplasma* antibody detection increased the detection to 98% of the cases [31]. Furthermore, quantitative analysis of antigen is helpful in monitoring response to therapy and assessing both prognosis and relapses [23,32,33,34].

The same galactomannan is detected in Histoplasmosis and Blastomycosis, leading to complete cross-reactivity. Cross-reactivity with other invasive fungal infections such as *Coccidioides* spp., *Aspergillus* spp., *Paracoccidioides* spp., *Sporothrix schenckii*, and *Talaromyces marneffei* can occur, though it can be differentiated by comparing concentrations [23,25,35,36,37,38].

### 3.3. Serology

Detection of anti-*H. capsulatum*-specific antibodies can aid in the diagnosis of certain histoplasmosis syndromes where antigen is unlikely to be positive (e.g., mediastinal granuloma, fibrosis) or increase the sensitivity in conjunction with antigen detection [1]. Major pitfalls include the time required for antibody production following an infection (4–8 weeks) and the inability of certain immunocompromised patients to mount a detectable humoral response [39]. The 3 most common antibody testing assays include complement fixation (CF), immunodiffusion (ID), and EIA [1].

A CF titer of 1:8 or higher is found in most patients with histoplasmosis, though titers of >1:32 or higher more strongly suggest active infections. In addition, an acute infection can be diagnosed by a ≥4-fold rise in antibody titers taken at least 2 weeks apart [1,20,26,39]. ID detects antibodies against the antigens H and M. The H band has poor sensitivity, being detected in only 20% of patients; however, its detection confirms acute infection. The M band can be detected in up to 80% of patients with histoplasmosis, though it is unreliable for the detection of ongoing or prior infection [39].

EIA-based antibody testing with IgG and IgM detection offers higher sensitivity than the ID and CF in patients with acute pulmonary histoplasmosis. In addition, combining antigen and EIA antibody testing provides an optimal method for the diagnosis of acute pulmonary histoplasmosis [40].

### 3.4. Molencular-Based Diagnostics

Molecular-based diagnostics rely on the detection of nucleic acid from a specific pathogen. There are a wide variety of laboratory-based polymerase chain reactions (PCR), though none are FDA-approved for clinical samples. The overall sensitivity of these tests ranges from 67% to 100% [41,42]. Currently, the main use of molecular-based diagnostics involved the use of the chemiluminescent DNA probe (AccuProbe test; Gen-Probe Incorporated, San Diego, CA, USA, 2011) from positive cultures [21].

Fungal sequencing assays targeting one or more regions in the multicopy ribosomal RNA (rRNA) genomic locus, such as the 18S rRNA, D1 and D2 regions of 28S rRNA, 5.8S rRNA, and internal transcribed spacers 1 and 2 (ITS1 and ITS2), have allowed for accurate and rapid identification of invasive fungal infections [43]. Further data are required to assess the performance of these tests.

More recently, next-generation sequencing (NGS), including the commercially available Karius test, has been shown to be able to detect over 400 different fungi [44]. This test relies on the detection of cell-free DNA in blood samples of patients with invasive fungal infections, including those with Histoplasmosis [45].

## 4. Clinical Presentation and Management

### 4.1. Pulmonary Histoplasmosis

#### 4.1.1. Acute and Subacute Pulmonary Histoplasmosis

In some highly endemic areas, 80–90% of the population will be exposed to histoplasmosis during their lifetime. However, over 90% of exposed individuals will have an unrecognized disease [46,47]. While less than 5% of those exposed to low-level inoculum develop symptoms, up to 75% of people will develop symptoms of infection following high-inoculum exposure. In addition, those exposed to intrinsically virulent strains or people with a weakened immune system may develop symptomatic disease with or without dissemination [26,48] (Table 2). Following 7 to 21 days from the inhalation of microconidia, the patients can experience flu-like symptoms, including fever, malaise, headache, weakness, dry cough, and chest discomfort. Improvement occurs promptly within 30 days in the majority of patients, though some symptoms may linger for months in some patients [25,49]. Acute pulmonary histoplasmosis is often confused with other diseases, mainly bacterial community-acquired pneumonia, leading to delays in the diagnosis. A recent study shows an average delay of 39.5 days prior to reaching the appropriate diagnosis of pulmonary histoplasmosis [50]. Newer rapid diagnostic techniques may lead to an earlier diagnosis. This may be particularly relevant to endemic areas in resource-scarce settings where other diagnostic techniques are not available. About 5% of the patients will develop acute rheumatological and/or dermatological manifestations, including erythema nodosum, erythema multiforme, and arthralgias/arthritis [51]. These reactions are related to hypersensitivity responses to *Histoplasma* antigens.

Chest imaging typically reveals diffuse patchy opacities or interstitial infiltrates, which can be associated with hilar and mediastinal adenopathy (Figure 4). Tree-in-bud opacities are less common. Nodules eventually calcify and are the most common finding from prior exposures [52]. Some individuals may develop large nodules, consolidations, acute respiratory distress syndrome, and progressive disseminated histoplasmosis. Of note, patients with disseminated disease often have normal chest imaging [29,53].

In mild-to-moderate cases, treatment is usually unnecessary, though Itraconazole (200 mg 3 times daily for 3 days and then 200 mg once or twice daily for 6–12 weeks) should be given to patients with symptoms lasting over 1 month [54].

In patients with moderately severe to severe acute pulmonary histoplasmosis, Lipid Amphotericin B (3.0–5.0 mg/kg daily intravenously for 1–2 weeks) followed by itraconazole (200 mg 3 times daily for 3 days and then 200 mg twice daily, for a total of 12 weeks) is recommended [54]. Recent studies have shown that liposomal amphotericin B accumulates in the reticuloendothelial system and other tissues for several weeks after administration of a dose, which opens the door for shorter durations of infusions [55]. However, this has not been tested for Histoplasmosis and is currently not recommended. Table 3.

#### 4.1.2. Pulmonary Nodules

Pulmonary nodules are a frequent finding in patients with resolved histoplasmosis and can be seen in up to 57% of the population in endemic areas with occupational exposures [56] (Figure 5). This nodule can occasionally contain no live organisms [57,58]. This poses a significant clinical challenge for the evaluation of patients in endemic regions. In a study of resected pulmonary granulomas of unknown cause, the most common identifiable etiology was histoplasmosis [59]. Rheumatological diseases, including sarcoidosis, may have a similar appearance and are part of the differential diagnosis [60]. Histoplasmosis must be excluded prior to the initiation of immunosuppressive therapy, as the use of corticosteroids can lead to progressive disseminated disease [61]. Most often, the nodules demonstrate several calcifications [57].

Given that the nodules may be similar to neoplastic processes, multiple studies have assessed whether newer imaging techniques can reliably distinguish a neoplastic process from a benign solitary pulmonary nodule. In a clinical study, FDG PET imaging was unable to distinguish between both entities and did not reduce the need for biopsies [62]. In another study, assessing the delayed radionuclide uptake by dual-time PET CT was able to discriminate between benign and malignant lung lesions with a positive predictive value of 80% and a negative predictive value of 100% [63]. *Histoplasma* serologies might be useful in differentiating between histoplasma nodules and lung cancers [64].

Management options for pulmonary nodules include serial follow-up CT imaging or invasive testing to assess for malignancy. For nodules <6 mm, annual screening with CT imaging may suffice. Nodules >8 mm may require closer follow-up (CT imaging at 3 months) or tissue sampling. The need for tissue sampling may vary according to the risk factors of the patient and nodule characteristics [65]. Ultimately, if no diagnosis is reached, an open lung biopsy frequently identifies a wide variety of infectious and inflammatory diseases [66]. Antifungal therapy is not required.

#### 4.1.3. Chronic Cavitary Pulmonary Histoplasmosis

Chronic cavitary pulmonary histoplasmosis (CCPH) can occur in 2% to 8% of patients with histoplasmosis. In a large outbreak of histoplasmosis in Indianapolis, older populations, male sex, white race, underlying immunosuppressive disorders, and chronic obstructive pulmonary disease were associated with this presentation [46]. Smoking has also been associated with chronic pulmonary histoplasmosis, likely due to its relationship with chronic lung disease [67]. Furthermore, it is unusual to occur in a patient without chronic lung disease.

Symptoms resemble those of pulmonary tuberculosis, with patients experiencing low-grade symptoms including cough, weight loss, anorexia, increased sputum production, fever/chills, night sweats, and hemoptysis. In older cohorts, when patients presented with advanced disease, upper lobe cavitation occurs almost universally (98%) and most often on the right apex (84% vs. 52%). Modern reports show that cavitations may occur in a much smaller percentage of cases (30%) [68]. These cavities tend to have a thick wall and may have apical pleural thickening (Figure 6). Most of the patients will also have evidence of prior calcified pulmonary granulomas, though the presence of hilar or mediastinal lymphadenopathy is rare.

Given the similarities with pulmonary tuberculosis, multiple cases of smear/culture/GeneXpert-negative “tuberculosis” in histoplasmosis-endemic areas have been subsequently found to have CCPH. In the 1950s, when tuberculosis was endemic in the United States, 7.2% of patients in the Missouri TB sanatorium were found to have CCPH [69]. It is likely that a significant number of patients with smear/culture/GeneXpert-negative cases of “tuberculosis” in histoplasmosis-endemic regions of developing countries are secondary to CCPH as diagnostic tools may be unavailable.

The diagnosis relies on the combination of clinical suspicion and laboratory data. Respiratory cultures are positive in 30–46% of the cases, though this can increase to up to 67% in bronchial washings [68,70,71]. Histoplasma antigen in BAL fluid [27] or urine can be detected in 87.5% of the patients. Serology is positive in 74–95% of the patients, and 75% of the patients have CF titers ≥ 1:16 [70,71].

All patients with CCPH should receive therapy, as therapy is associated with decreased mortality, and regression of the pulmonary infiltrates in two-thirds of cases [72]. The current guidelines recommend itraconazole (200 mg 3 times daily for 3 days and then once or twice daily for at least 1 year), but some experts recommend 18–24 months in view of the risk for relapse, which occurs in 10% to 20% of the cases [54].

## 5. Mediastinal Histoplasmosis

### 5.1. Mediastinal Lymphadenopathy (or Mediastinal Adenitis)

In endemic areas, children or young adults presenting with fever and an enlarged mediastinal cavity are usually related to acute histoplasmosis. The diagnosis is usually made with radiography in conjunction with the clinical picture and non-invasive testing for acute pulmonary histoplasmosis. While it is usually self-resolving, some patients may develop symptoms related to the encroachment of adjacent structures, including the airways, esophagus, or superior vena cava [29] (Table 4).

Non-steroidal anti-inflammatory treatment can be given to those with significant symptomatology. Antifungal therapy is usually not indicated as there is no evidence that it reduces the risk of progression to other syndromes, though it should be considered if the patient is receiving steroids [1].

### 5.2. Mediastinal Granuloma

Mediastinal granuloma is the abnormal enlargement of mediastinal lymph nodes due to granulomatous inflammation, which can be caseating or non-caseating. The coalesced group of lymph nodes usually has a thin fibrotic capsule. Patients with thin capsules (thickness ≤ 5 mm) usually do not have any significant symptomatology, though thicker fibrotic capsules (greater than 10 mm) can lead to invasion of adjacent structures [73]. The mass is usually calcified in a diffuse or subcapsular pattern. The core can show granulomas or necrotic tissue. On occasion, yeast can be found on histopathological examination, though it rarely grows in cultures [73]. While serologies are usually positive, antigen examination in plasma or urine is negative.

Mediastinal granuloma usually involves the right paratracheal nodes, while mediastinal fibrosis usually involves the subcarinal area. The clinical manifestations are related to the anatomical location of the granuloma. Right paratracheal involvement is associated with superior vena cava obstruction or azygous vein involvement, while involvement of the hilar lymph nodes is associated with compression of the pulmonary vessels and bronchi. A mass effect on the esophagus may be associated with dysphagia [26].

Therapy, with or without steroids, is usually not required, though it can be considered in symptomatic patients. Current guidelines recommend itraconazole, though there are no controlled trials to prove its efficacy, and there is no consensus on how long therapy should be continued.

### 5.3. Fibrosing Mediastinitis

Fibrosing mediastinitis or mediastinal fibrosis is characterized by the encasement of mediastinal structures by invasive fibrotic tissue thought to be secondary to an exuberant and abnormal response to a prior infection. It was hypothesized to be a progression from mediastinal granuloma, though recent reports do not show an association between mediastinal granuloma and fibrosing mediastinitis, suggesting that they are independent syndromes [74]. While mediastinal granulomas occur in all age groups, fibrosing mediastinitis typically occurs in patients aged 20–40 years old [29]. In a study of 19 cases of mediastinal fibrosis, the presence of HLA-A2 was strongly associated with mediastinal fibrosis, suggesting an abnormal immune response [75]. Another study found that HLA-DQB1*04:02 was associated with mediastinal fibrosis, further supporting the hypothesis of an aberrant immune response [76].

While the progression of disease occurs slowly, symptoms occur rather acutely in late presentations once the fibrosis impinges on the mediastinal structures, such as the esophagus, airway, and great vessels past a critical point [77]. The most common presentation is hemoptysis [74]. Other symptoms include dyspnea, chest pain, dysphagia, signs of superior vena cava syndrome, and heart failure. Most often, the impingement is unilateral, and cases of autoamputation of one lung have been reported as a consequence of right pulmonary artery occlusion. Systemic symptoms of infection, such as fever, chills, and night sweats, are absent.

Chest X-rays can be deceptive as subcarinal masses may hide behind the cardiac silhouette. In contrast, chest CT reveals abnormal mediastinal proliferation with heavy calcification, usually localized at the lymph node sites (Figure 7). A FDG positron emission tomography (PET) scan shows intense FDG avidity in the mediastinal tissue.

Histopathology is characterized by the presence of mixed, lymphocytic inflammatory infiltrates. The infiltrates display a large proportion of CD-20 positive B lymphocytes, forming a peripheral rim surrounding the fibrotic lesion. In some cases, cytotoxic T cells are found within the fibrotic area. Given the presence of CD-20 positive B lymphocytes and persistent inflammation noted on PET CT scans, B-cell depletion has been investigated as a potential therapeutic option [78]. Recently, four weekly rituximab infusions with itraconazole led to a reduction in fibroinflammatory changes noted on PET CT and overall clinical improvement in 3 patients with mediastinal fibrosis [79].

Other therapeutic strategies have focused on the use of stents to improve anatomical impingement or bronchial artery embolization in patients with hemoptysis. Surgical debridement and pneumectomies are high-risk procedures with a mortality rate of up to 20%. Antifungal therapy has no role in the management of mediastinal fibrosis.

## 6. Progressive Disseminated Histoplasmosis

Progressive disseminated histoplasmosis is defined as a clinical illness that does not improve after at least 3 weeks of observation and that is associated with physical or radiographic findings and/or laboratory evidence of involvement of extrapulmonary tissues [54]. Immunocompromised patients have at least 10 times higher chances of developing progressive disseminated histoplasmosis when compared with their immunocompetent counterparts. Infancy and ages greater than 55 also predispose patients to PHD. It was typically seen in patients with advanced HIV and remains a major presenting disease in patients with advanced HIV in resource-limited settings. It has also been seen in patients with CD4 lymphopenia, common variable immunodeficiency, INF-γ, TNF-α and IL-12 pathway deficiencies, hyper-IgE syndrome, and iatrogenic immunosuppression, among others. Recipients of anti-TNF-α therapy and solid organ transplants are an increasing at-risk population to PDH.

Symptoms are usually non-specific and may include fever, fatigue, and weight loss. Pulmonary symptoms may not be predominant and can include a dry cough and shortness of breath. Pulmonary imaging can show diffuse miliary-type reticulonodular infiltrates (Figure 8). Patients may present with hepatosplenomegaly, extrapulmonary lymphadenopathy, CNS lesions, and skin lesions. Gastrointestinal involvement is common and may mimic a cytomegalovirus infection. It can present with diarrhea, mucosal thickening, and ulcerations. Laboratory analysis may show cytopenia, hepatic enzyme elevation, and high lactate dehydrogenase.

PDH can also present with severe electrolyte imbalance, which may be secondary to involvement of the adrenal glands [80]. PDH is a rare cause of 1,25-dihydroxy vitamin-D-mediated hypercalcemia [81].

Some patients may develop hemophagocytic lymphocytosis syndrome, macrophage activation syndrome, septic shock, or acute hypoxemic respiratory failure. Hence, therapy is required for all patients with PDH.

Patients with severe disease should receive intravenous Liposomal Amphotericin B until they are stabilized (usually 1 to 2 weeks), followed by itraconazole monotherapy. Liposomal formulation is preferred over deoxycholate as it has been shown to have higher rates of response (88% vs. 64%) and is associated with fewer side effects [82]. Upon completion of the initial therapy with amphotericin B, patients should be transitioned to oral itraconazole (200 mg 3 times daily for 3 days and then 200 mg twice daily) for a total of at least 12 months [83]. Itraconazole suspension has a more reliable pharmacokinetic (PK) than itraconazole capsules, though it is associated with increased gastrointestinal side effects. Severe acute respiratory syndrome can occur following the initiation of therapy (paradoxical reactions) or after the restoration of host immunity (immune reconstitution syndrome) [84]. Patients who develop severe respiratory complications should receive methylprednisolone (0.5–1 mg/Kg IV daily) for the first 1–2 weeks [54].

In patients with mild-to-moderate disease, therapy may be with itraconazole monotherapy and should be continued for at least 1 year as well and until all clinical findings have resolved and Histoplasma antigen, if initially positive, has reverted to negative or is below 2 ng/mL. Patients should have antigen levels measured throughout the therapy. Low-level antigen may persist for years following the completion of therapy [54].

## 7. Other Considerations: Therapeutic Drug Monitoring

Therapeutic drug monitoring is indicated when the standard dosing of any given drug results in marked variability in drug blood concentrations and when relationships between drug exposure (plasma drug level) and either efficacy or toxicity have been established. The plasma drug levels of multiple azoles vary due to inconsistent absorption, metabolism, elimination, or interaction with concomitant medications [85,86,87].

While Isavuconazole has predictable (linear) pharmacokinetics (PK) with a direct correlation between dosing and blood levels, Itraconazole and posaconazole have variable pharmacokinetics, leading to significant PK variability despite the recommended therapeutic dosing ranges. For that reason, blood levels of itraconazole and posaconazole should be monitored throughout the therapy. There is ample evidence that therapeutic drug monitoring is associated with better treatment success and less drug discontinuation due to adverse events [88,89].

Itraconazole has suboptimal absorption and is particularly poor under circumstances of impaired GI integrity, a reduced acidic environment, which is required for capsules, and oral intolerance leading to poor compliance. This has been improved with newer formulations. For itraconazole, when measured by high-pressure liquid chromatography (HPLC), both itraconazole and its bioactive hydroxy-itraconazole metabolite are reported, the sum of which should be considered in assessing drug levels, and it should be at least 2 mcg/mL [86,90].

New antifungal classes have been under development and are in the last stages of clinical testing. Fosmanogepix, Olorofim, and Ibrexafungerp have potent activity against *H. capsulatum* and may be helpful for those who are unable to tolerate Itraconazole or other azoles [91].

## 8. Special Populations

### 8.1. Solid Organ Transplant Recipients

While histoplasmosis accounts for <5% of all invasive fungal infections in solid organ transplant recipients, the infections can be devastating [92,93]. In a large study in an endemic area, histoplasmosis (proven or probable) occurred in 14 out of 3436 transplant recipients during a 10-year period [94].

Transplant-associated histoplasmosis can occur as a de-novo infection from an environmental exposure, as the reactivation of a latent infection or as a donor-derived transmission [95]. Most of the cases occur within the first 2 years following transplantation, though the median time from transplantation varies among the different studies [92,94,95,96]. Donor-derived histoplasmosis may occur in 1:10,000 transplants and is characterized by occurring in the early post-transplant period [83].

Pre-transplant serologies or chest imaging consistent with prior exposure to histoplasmosis are not associated with post-transplant histoplasmosis. Hence, post-transplant primary prophylaxis is not recommended in previously exposed patients [97]. However, secondary prophylaxis should be considered in patients who have recovered from active histoplasmosis during the 2 years before the initiation of immunosuppression [98].

### 8.2. Hematological Malignancies and Stem Cell Transplant Recipients

Despite their profound immunosuppressive states, PDH has been rarely described in stem cell transplant recipients. It is unclear if this may be related to the use of antifungal prophylaxis or underreporting. Most cases occurred following therapy with steroids for graft versus host disease [99,100]. Currently, monitoring antigen levels or initiating primary prophylaxis in patients living in endemic areas is not recommended [54].

Patients with hematological malignancies, including chronic lymphocytic leukemia, are at increased risk of PDH. In these patients, PDH can mimic the rapid progression of disease (Richter’s transformation) [101]. It is important for clinicians to maintain a high level of suspicion in immunocompromised patients living in endemic areas.

### 8.3. Human Immunodeficiency

Histoplasmosis occurs in 2–5% of patients with HIV/AIDS in endemic areas [102]. In Latin America, the estimated incidence of histoplasmosis in 2012 was 1.48 cases per 100 people, though this is likely to be an underestimate of the cases as many are misdiagnosed as tuberculosis [8]. In certain areas, it can represent up to 50–75% of the AIDS-defining infections [103]. Most cases are associated with a CD4 T lymphocyte (CD4) count <150 cells/mm^3^. Over 95% of the patients will present with progressive disseminated disease, with mortality rates ranging between 10% and 60% [102,103].

Therapy should be similar to that for other immunocompromised groups and depend on the severity of the disease. Severe disease should be treated with 1–2 weeks of induction therapy with amphotericin, followed by monotherapy with itraconazole for 12 months. Less than 12 months can be considered in patients who are clinically stable and have achieved immune reconstitution. Patients who are unable to achieve immune reconstitution may require longer than 12 months of therapy. Antiretroviral therapy should be initiated as soon as possible in those who are not suspected to have central nervous system involvement [104].

### 8.4. Biological and Small Molecule Targeted Immunomodulatory Therapies

While multiple types of immunosuppressive medications may be associated with an increased risk of histoplasmosis, patients receiving TNF-α inhibitors (infliximab, etanercept, adalimumab, certolizumab, golimumab) are at the highest risk of intracellular infections, including histoplasmosis. Blockage of TNF-α inhibits macrophage fungistatic activity [105]. The incidence of histoplasmosis in patients with infliximab has been estimated to be 18.75 per 100,000 persons, while the incidence with etanercept has been estimated at 2.65 per 100,000 persons. TNF-α inhibitors should be stopped and should only be resumed once the patient has completed therapy and all signs of residual disease have resolved. If therapy is resumed, then itraconazole should be continued as long as the patient is on the inhibitor medication [106,107].

Bruton tyrosine kinase inhibitors (BTK), acalabrutinib, ibrutinib, and zanubrutinib, have been associated with increased risk for invasive fungal infections. A retrospective study showed that 4.1% of patients developed an opportunistic infection, 74% of which were invasive fungal infections. Other molecules associated with increased risk of histoplasmosis include Janus kinase inhibitors and phosphatidylinositol-3-kinase (PI3K) inhibitors. Lymphodepleting agents, in particular alemtuzumab, have also been associated with an increased risk of histoplasmosis [108].

## 9. Summary

*Histoplasma capsulatum*, the etiological agent for histoplasmosis, is a dimorphic fungus that grows as a mold in the environment and as a yeast in human tissues. It has a broad global distribution with shifting epidemiology during the recent decades. The clinical presentations and outcomes of infection depend on the dynamic interactions between the host’s innate and adaptive immunity and the fungal virulence factors. Further research is required to develop less toxic treatment regimens and to tailor the length of therapy in immunocompromised host.

## Figures and Tables

**Figure 1 jof-09-00236-f001:**
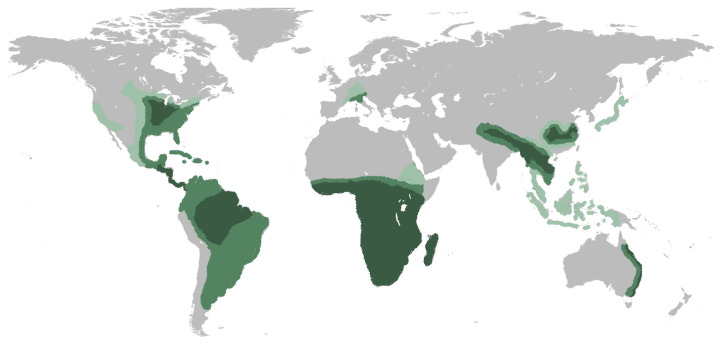
World map epidemiology of Histoplasmosis. Dark green represents areas likely to be hyperendemic, green represents areas where infection occurs regularly. Light green represents areas where local infections have been reported.

**Figure 2 jof-09-00236-f002:**
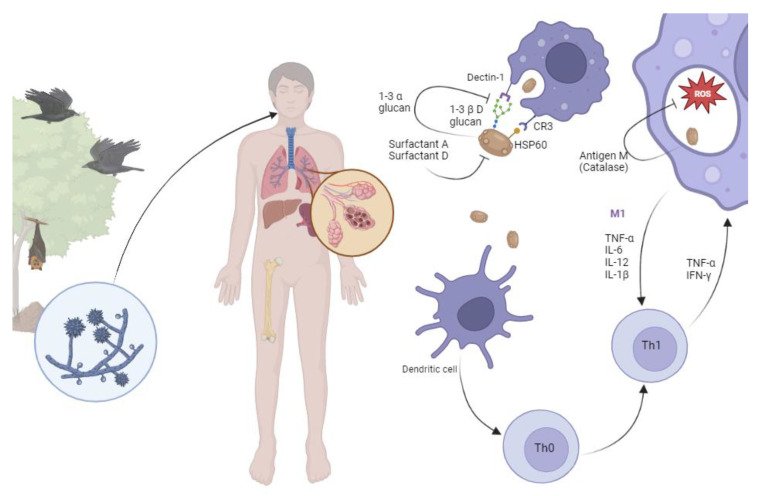
Pathogenesis of histoplasmosis. Aerosolized microconidia are inhaled by the host. Host temperature (37 °C) triggers morphological transformation to yeast. Surfactant A and D (collectins) display a direct fungicidal role through a calcium-dependent mechanism of yeast permeabilization. Heat shock protein 60 (HSP60) is recognized by complement receptor 3 (CR3) and promotes phagocytosis. If Dectin-1 is activated by interaction with 1-3 β D glucan, the macrophage is able to produce a profound inflammatory cascade. To avoid it, the yeast covers the 1-3 β D glucan with 1-3 α glucan. Once in the phagolysosome, antigen M inhibits the reactive oxygen species (ROS). The infected macrophage can migrate to any other organ in the body, including liver, spleen, and bone marrow. Dendritic cells are able to kill the yeast and present antigens to T helper cells (Th0) and promote their polarization into Th1, which in turn increases pro-inflammatory cytokines leading to further macrophage activation.

**Figure 3 jof-09-00236-f003:**
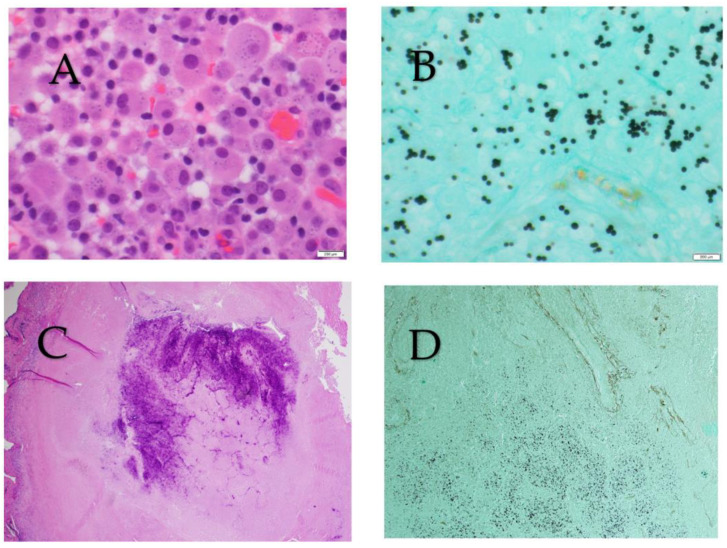
(**A**) Giemsa stain showing intracellular small ovoid yeast. (**B**) Grocott-Gomori’s methenamine silver showing small ovoid, narrow-based budding yeasts. (**C**) Hematoxylin-eosin stain of a caseating pulmonary granuloma. (**D**) Grocott-Gomori’s methenamine silver showing small yeast in a pulmonary granuloma.

**Figure 4 jof-09-00236-f004:**
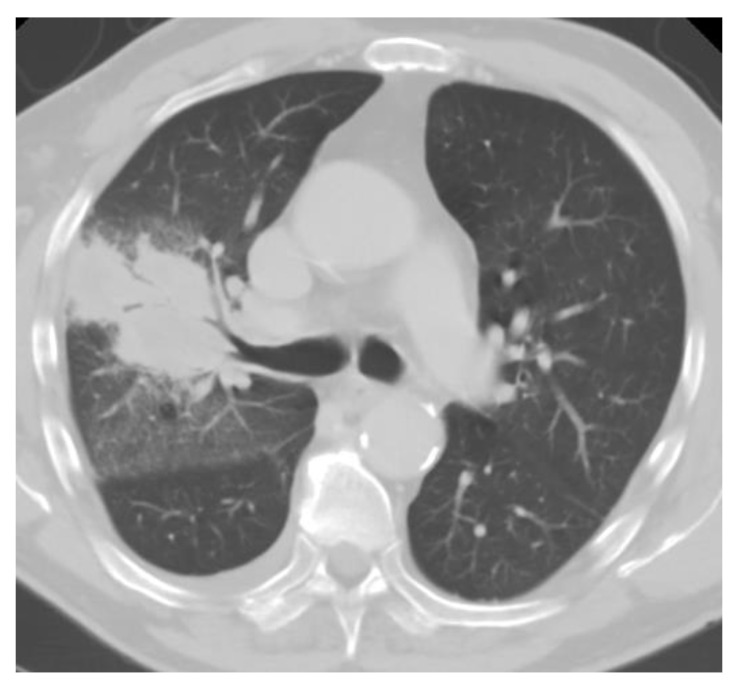
Chest CT of acute pulmonary histoplasmosis showing a large consolidation with air bronchogram.

**Figure 5 jof-09-00236-f005:**
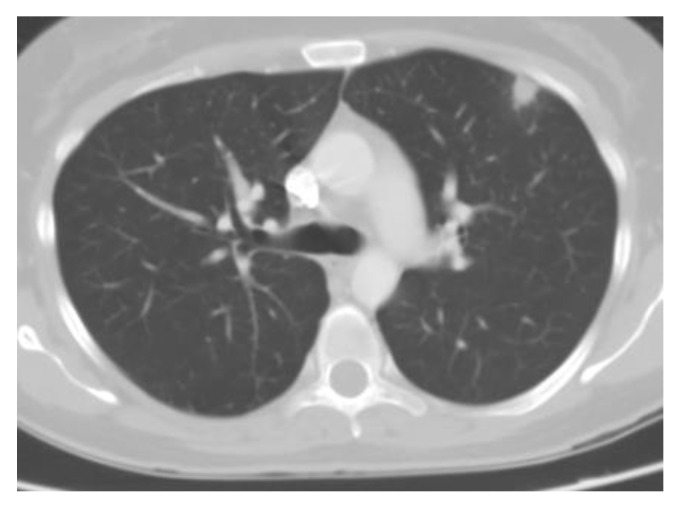
Incidental pulmonary nodule in a patient from a highly endemic area. Biopsies often reveal granulomatous changes and calcifications.

**Figure 6 jof-09-00236-f006:**
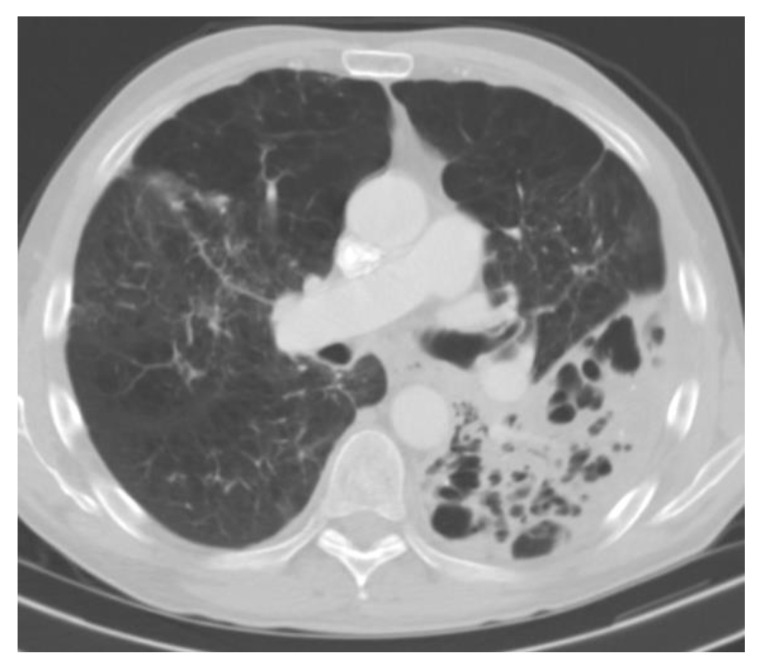
Chronic cavitary pulmonary histoplasmois. Multiple thick wall cavitations in a patient with emphysematous disease.

**Figure 7 jof-09-00236-f007:**
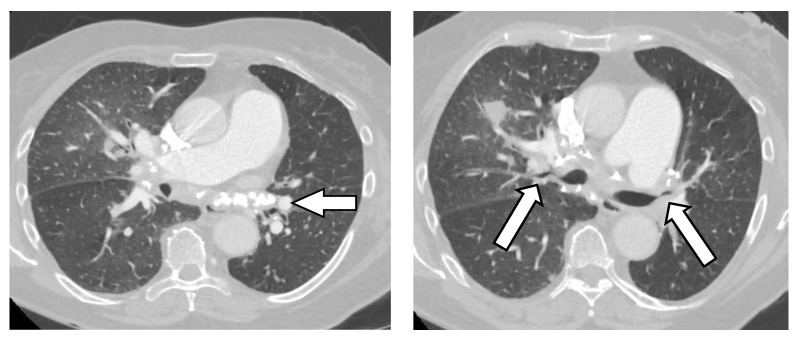
Fibrosing mediastinitis. (**Left** image) Arrows points at heavy calcification in the mediastinum. (**Right** image) both arrows point at severe stenosis of bronchial lumen.

**Figure 8 jof-09-00236-f008:**
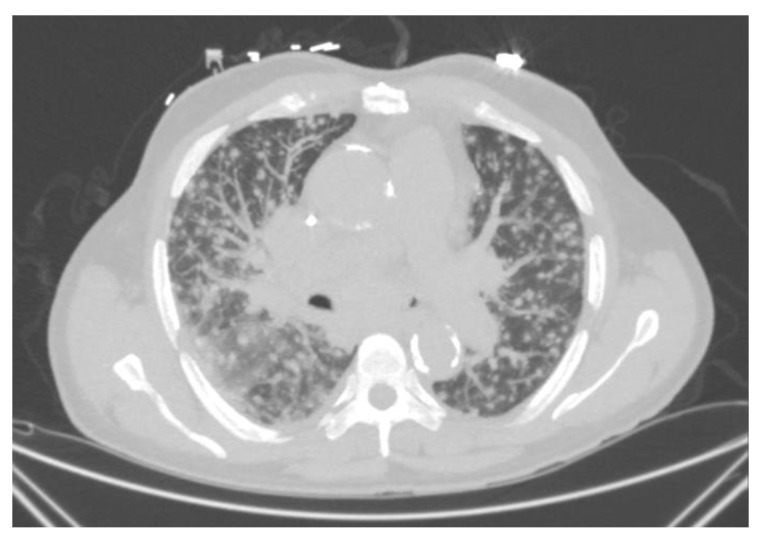
Progressive disseminated histoplasmosis. CT chest showing diffuse reticulonodular infiltrates with large mediastinal lymphadenopathies.

**Table 2 jof-09-00236-t002:** Key features of pulmonary histoplasmosis.

Clinical Form				
Characteristic	Acute	Subacute	Nodular	Chronic Cavitary
Age	Any	Any	Any	>50-year-old with structural lung disease
Clinical manifestation	Fever, headache, dry cough, chills, chest pain, malaise, myalgias and arthritis	Same as acute but symptoms are milder	Usually asymptomatic	Fever, productive cough, dyspnea, weight loss, hemoptysis, night sweats, chest pain
Symptom duration	1–2 weeks	Weeks to months	-	Months to years
Mimicked disease	Community acquired pneumonia	Community acquired pneumonia	Neoplasm	Tuberculosis, Sarcoidosis
Pathology	Granuloma with acute lung injury	Well-formed granulomas	Well-formed granulomas	Cavities with granulomas, tissue destruction
Radiologic findings	Diffuse bilateral patchy opacities	Focal or patchy opacities	Nodules	Cavitation, fibrosis, volume loss, pleural thickening. Right upper lobe is most commonly affected
Hilar and Mediastinal lymph nodes	Enlarged	Enlarged	Not enlarged	Not enlarged. Occasionally calcified
Calcifications	None	None	Present	Present
Indications for treatment	Severe disease	Symptoms over 1 month	None	Yes

**Table 3 jof-09-00236-t003:** Clinical forms and treatment recommendations.

Clinical Form	Treatment Recommendation
Pulmonary	
Acute—Mild to moderate	
Immunocompetent host	
<4 weeks	Usually unnecessary
>4 weeks	Itraconazole for 6–12 weeks
Immunocompromised host	
Regardless of duration	Itraconazole for 12 months
Acute—Moderately severe or severe	
Immunocompetent host	Lipid Amphotericin B for 1–2 weeks followed by Itraconazole for 12 weeks
Immunocompromised host	Lipid Amphotericin B for 1–2 weeks followed by Itraconazole for at least 12
	months and negative or low antigen (<2 ng/mL)
	Methylprednisolone 0.5–1 mg/Kg during the first 1–2 weeks if the patient
	develops ARDS
Subacute	Itraconazole for 6–12 weeks
Nodular	None
Chronic cavitary	Itraconazole for at least 12 months
**Mediastinal**	
Adenitis	As acute pulmonary
Granuloma	
Asymptomatic	None
Symptomatic	Itraconazole for 6–12 weeks
Fibrosis	Symptomatic management (e.g., stents)
	Antifungal therapy not recommended
	Can consider Rituximab in certain cases
**Progressive disseminated**	
Mild to moderate	
Immunocompetent host	Itraconazole for 6–12 weeks
Immunocompromised host	Itraconazole for 12 months
Moderately severe or severe	
Immunocompetent host	Lipid Amphotericin B for 1–2 weeks followed by Itraconazole for 12 weeks
Immunocompromised host	Lipid Amphotericin B for 1–2 weeks followed by Itraconazole for at least 12
	months and negative or low antigen (<2 ng/mL)
	Methylprednisolone 0.5–1 mg/Kg during the first 1–2 weeks if the patient
	develops ARDS

**Table 4 jof-09-00236-t004:** Key features of mediastinal histoplasmosis.

Clinical Form			
Characteristic	Adenitis	Granulomatous	Fibrosing
Age	Usually < 20 y	All ages > 2 y	Typically 20–30 y
Clinical manifestation	Usually detected during diagnosis of acute pulmonary histoplasmosis. Can have mild obstructive symptoms	Obstructive syndromes (e.g., SVC, dysphagia and chest pain can occur)	Obstructive syndromes (e.g., SVC, pulmonary artery veins, dysphagia, dyspnea)
Pathology	Granulomas	Granulomas with extensive necrosis	Extensive fibrosis with or without granulomas
Radiologic findings	Large nodes, not calcified	Large mass, subcapsular or diffuse calcifications	Proliferative calcified mediastinal nodes with obstruction
Calcifications	None	Usually	Extensive
Indication for treatment	As acute pulmonary	Obstruction or pain	Recurrent hemoptysis, obstruction

## Data Availability

Not applicable.

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
