# Peer review of "Pulmonary Histoplasmosis: A Clinical Update"

_jof, 2023, doi:10.3390/jof9020236_

Round 1
Reviewer 1 Report
J of F 2121011
HISTOPLASMOSIS
This is an excellent review written by leading authorities on the subject. I have some rather minor points for the authors to consider for their revision
Diagnosis: Consider emphasizing the fact that delays in diagnosis ( Typically > 1 month) are not uncommon (see Frequency and Duration of, and Risk Factors for, Diagnostic Delays Associated with Histoplasmosis. Miller AC et al. J Fungi (Basel). 2022 Apr 23;8(5):438. doi: 10.3390/jof8050438.J Fungi (Basel). 2022. PMID: 35628693). Would the development of point of care diagnostics aid in the diagnosis of pulmonary histoplasmosis among “ community-acquired” pneumonias?
Table 1 is very nice, but can the authors put an indication from which references they used the percentages shown?
Consider emphasizing some unique association of disseminated histoplasmosis, that of adrenal involvement and the rare cause of 1, 25-dihydroxy vitamin-D-mediated hypercalcemia
Consider emphasizing the pulmonary histoplasmosis is not an uncommon simulator of lung cancer (see Eur J Clin Microbiol Infect Dis . 2013. Jan;32(1):101-5. doi: 10.1007/s10096-012-1720-9. Epub 2012 Aug 16. Open-lung biopsy in patients with undiagnosed lung lesions referred at a tertiary cancer center is safe and reveals noncancerous, noninfectious entities as the most common diagnoses)
L 491: consider clarifying that itraconazole suspension is preferred compared to tablets, because of more reliable PKs)
L 503. Could add a paragraph of the potential role of investigational agents currently undergoing clincal testing (see eg The Antifungal Pipeline: Fosmanogepix, Ibrexafungerp, Olorofim, Opelconazole, and Rezafungin. Hoenigl M et al.. Drugs. 2021 Oct;81(15):1703-1729. doi: 10.1007/s40265-021-01611-0. Epub 2021 Oct 9.)
549 Consider emphasizing the high risk of lymphoid malignancies, such is CLL, with histoplasmosis and the fact that histoplasmosis can simulate Richter’s transformation ( CLL progression) see Disseminated histoplasmosis mimicking relapsed chronic lymphocytic leukaemia. .Shahani L. BMJ Case Rep. 2018 Jun 28;2018)
Author Response
[R1] This is an excellent review written by leading authorities on the subject. I have some rather minor points for the authors to consider for their revision
[Authors] We appreciate the kind words.
[R1] Diagnosis: Consider emphasizing the fact that delays in diagnosis ( Typically > 1 month) are not uncommon (see Frequency and Duration of, and Risk Factors for, Diagnostic Delays Associated with Histoplasmosis. Miller AC et al. J Fungi (Basel). 2022 Apr 23;8(5):438. doi: 10.3390/jof8050438.J Fungi (Basel). 2022. PMID: 35628693). Would the development of point of care diagnostics aid in the diagnosis of pulmonary histoplasmosis among “community-acquired” pneumonias?
[Authors] We have added the following paragraph on lines 289-294:
“Acute pulmonary histoplasmosis is often confused with other diseases, mainly bacterial community acquired pneumonia, leading to delays in the diagnosis. A recent study shows an average delay of 39.5 days prior to reaching the appropriate diagnosis of pulmonary histoplasmosis. Newer rapid diagnostic techniques may led to earlier diagnosis. This may be particularly relevant to endemic areas in resource-scarce settings where other diagnostic techniques are not available.”
[R1] Table 1 is very nice, but can the authors put an indication from which references they used the percentages shown?
[Authors]: We have added the reference.
[R1] Consider emphasizing some unique association of disseminated histoplasmosis, that of adrenal involvement and the rare cause of 1, 25-dihydroxy vitamin-D-mediated hypercalcemia
[Authors]: We have added the following paragraph on lines 519-521:
PDH can also present with severe electrolyte imbalance which may be secondary to involvement of the adrenal glands. PDH is a rare cause of 1, 25-dihydroxy vita-min-D-mediated hypercalcemia.
[R1] Consider emphasizing the pulmonary histoplasmosis is not an uncommon simulator of lung cancer (see Eur J Clin Microbiol Infect Dis . 2013. Jan;32(1):101-5. doi: 10.1007/s10096-012-1720-9. Epub 2012 Aug 16. Open-lung biopsy in patients with undiagnosed lung lesions referred at a tertiary cancer center is safe and reveals noncancerous, noninfectious entities as the most common diagnoses)
[Authors] Line 340-347 describes the similarities between histoplasma pulmonary nodules and lung cancer. We have added the following sentence on line 353-355 with the reference provided:
Ultimately, if no diagnosis is reached, open lung biopsy frequently identifies a wide variety of infectious and inflammatory diseases.
[R1] L 491: consider clarifying that itraconazole suspension is preferred compared to tablets, because of more reliable PKs)
[Authors]: Added on line 532-533. Itraconazole suspension has a more reliable pharmacokinetic (PK) than itraconazole capsules, though it is associated with increased gastrointestinal side effects.
[R1] L 503. Could add a paragraph of the potential role of investigational agents currently undergoing clincal testing (see eg The Antifungal Pipeline: Fosmanogepix, Ibrexafungerp, Olorofim, Opelconazole, and Rezafungin. Hoenigl M et al.. Drugs. 2021 Oct;81(15):1703-1729. doi: 10.1007/s40265-021-01611-0. Epub 2021 Oct 9.)
[Authors]: We have added the following paragraph on line 568-571:
New antifungal classes have been under development and are within the last stages of clinical testing. Fosmanogepix, Olorofim and Ibrexafungerp have potent activity against H. capsulatum and may be helpful in those who are unable to tolerate Itraconazole or other azoles
[R1] 549 Consider emphasizing the high risk of lymphoid malignancies, such is CLL, with histoplasmosis and the fact that histoplasmosis can simulate Richter’s transformation (CLL progression) see Disseminated histoplasmosis mimicking relapsed chronic lymphocytic leukaemia. Shahani L. BMJ Case Rep. 2018 Jun 28;2018)
[Authors] We have added the following paragraph on line 604-607:
Patients with hematological malignancies, including chronic lymphocytic leukemia are at increased risk of PDH. In these patients, PDH can mimic rapid progression of disease (Richter’s transformation). It is important for clinicians to maintain a high level of suspicion in immunocompromised patients living in endemic areas.
Reviewer 2 Report
It is well written, with major emphasis on serological assays to diagnosis and follow, and with extensive references. I think it is an excellent paper that should be accepted without major revisions Adding some Figures to include the histological features of Histo would be useful. Also, it would be nice to include some examples of CT scans showing acute pulmonary histo, chronic forms, adenopathy which many clinicians reading the article would probably enjoy.
Author Response
It is well written, with major emphasis on serological assays to diagnosis and follow, and with extensive references. I think it is an excellent paper that should be accepted without major revisions Adding some Figures to include the histological features of Histo would be useful. Also, it would be nice to include some examples of CT scans showing acute pulmonary histo, chronic forms, adenopathy which many clinicians reading the article would probably enjoy.
[Authors]: We have added images:
- Figure 3: Histopathology of Histoplasmosis showing intracellular yeasts and small narrow-based budding yeasts.
- Figure 4: CT chest of acute pulmonary histoplasmosis
- Figure 5: Incidental pulmonary nodule
- Figure 6: Chronic cavitary pulmonary histoplasmosis
- Figure 7: fibrosing mediastinitis with extensive calcifications and stenosis of bronchial lumen.
- Figure 8. Progressive disseminated histoplasmosis with reticulonodular infiltrates and lymphadenopathy.